# A Comparative Analysis of Conventional and Deep Eutectic Solvent (DES)-Mediated Strategies for the Extraction of Chitin from Marine Crustacean Shells

**DOI:** 10.3390/molecules26247603

**Published:** 2021-12-15

**Authors:** Kellie Morgan, Colin Conway, Sheila Faherty, Cormac Quigley

**Affiliations:** Galway-Mayo Institute of Technology, Dublin Road, H91 T8NW Galway, Ireland; Kellie.Morgan@gmit.ie (K.M.); colin.conway@gmit.ie (C.C.); sheila.faherty@gmit.ie (S.F.)

**Keywords:** chitin, chemical extraction, deep eutectic solvent (DES), crustacean shells, green chemistry, green solvents, marine waste processing, crab, shrimp, lobster

## Abstract

Chitin, the second most abundant biopolymer on earth, is utilised in a wide range of applications including wastewater treatment, drug delivery, wound healing, tissue engineering, and stem cell technology among others. This review compares the most prevalent strategies for the extraction of chitin from crustacean sources including chemical methods that involve the use of harsh solvents and emerging methods using deep eutectic solvents (DES). In recent years, a significant amount of research has been carried out to identify and develop environmentally friendly processes which might facilitate the replacement of problematic chemicals utilised in conventional chemical extraction strategies with DES. This article provides an overview of different experimental parameters used in the DES-mediated extraction of chitin while also comparing the purity and yields of associated extracts with conventional methods. As part of this review, we compare the relative proportions of chitin and extraneous materials in different marine crustaceans. We show the importance of the species of crustacean shell in relation to chitin purity and discuss the significance of varying process parameters associated with different extraction strategies. The review also describes some recent applications associated with chitin. Following on from this review, we suggest recommendations for further investigation into chitin extraction, especially for experimental research pertaining to the enhancement of the “environmentally friendly” nature of the process. It is hoped that this article will provide researchers with a platform to better understand the benefits and limitations of DES-mediated extractions thereby further promoting knowledge in this area.

## 1. Introduction

Chitin is a natural polysaccharide biopolymer found in crustacean shells, insect exoskeletons, and some fungi and algae [1]. Next to cellulose, it is the second most abundant biopolymer on earth [2]. The important properties normally associated with chitin include biodegradability, biocompatibility, bioactivity, non-toxicity, and wound healing. Chitin is currently used in biomedicine, cosmetics, agriculture, and water treatment industries [3]. Furthermore, chitin can be derivatised into more soluble products such as chitosan; chitosan exhibits potential in the food, environmental protection, and biomedicine industries [4].

Chitin is white in colour, insoluble in water, inelastic, and commercially produced from crustacean exoskeletons. Chitin constitutes approximately 15–40% of crustacean shells; 20–50% of the shell is composed of calcium carbonate (CaCO_3_) and 20–40% of protein [5]. Removal of the latter materials is typically required to extract chitin from crustacean shells [1]. As such, the term extraction is counterintuitive; in fact, it is all of the other materials which are extracted, and the purified chitin remains as a solid following the removal of all the other components in the substrate material. Saravana et al. [2] reported the extraction of chitin with less than 2% contaminants (CaCO_3_ and protein) when chitin was extracted from shrimp shells (*Marsupenaeus japonicus*) using a conventional extraction approach. This shell type (*Marsupenaeus japonicus*) contains 19.21% chitin, and a yield of 16.08% (by weight of the original shell) was achieved in the latter study, indicating that some chitin was lost during the process. The conventional method for extracting chitin involves the use of harsh solvents, causing environmental problems due to the generation of harsh acid–base wastewater [2]. Optimally, the method to extract chitin would adopt green chemistry principles, using safer solvents for the extraction of CaCO_3_ and protein, culminating in maximal chitin yields and purity.

Deep eutectic solvents (DES) were first identified in 2001 by Abbott et al. [6] and have since been applied in fields such as material synthesis, biotransformation, and extraction [7]. According to Torregrosa-Crespo et al. [8], approximately 800 papers have since been published on DES. Recent studies [1,2,3,4,9] have shown that DES can be used as alternatives to harsh solvents during chitin extraction from crustacean shells. A green chemistry approach for chitin purification using DES offers many advantages: raw materials which are abundant and inexpensive, safer solvents, potential to reuse solvents, lower toxicity, higher biodegradability, and negligible volatility [10]. To be considered a viable alternative to the chemical method, DES-mediated extractions should produce chitin of similar or higher purity.

This review utilised documentary analysis to compare the conventional method of chitin extraction from crustacean shells with DES-mediated extraction procedures based upon the purity of the chitin produced. The methods used by the different studies to determine chitin purity will initially be discussed. An analysis of the considerations associated with higher purity and relevant trends in the literature were analysed to compare the DES-mediated method with the conventional method. Factors including temperature, reaction time, and the species of crustacean shell utilised will be considered. Furthermore, additional criteria such as yield, the quality of the chitin extracted, the efficiency of the process, the environmental impact, and the cost of the processes will also be discussed. To the best of the authors’ knowledge, this is the first review to compare DES-mediated chitin extraction from crustacean shells to the conventional method in terms of chitin purity.

## 2. Methodology

PubMed search criteria included publication dates from 2002 until October 2021. An all-field search for “extraction” AND “crustacean shells”, with MeSH terms “chitin” and all-field search of “demineralization”, and their combinations, selected 30 articles. An additional search for MeSH “chitin” and all-field “deep eutectic solvents” collected 15 articles. Of these 45 articles, 16 were excluded as they utilised fermentation methods for chitin purification; 29 articles were scrutinised for the review. Additional articles were found after scrutinising the references of selected papers.

## 3. Chitin

### 3.1. Chitin Properties

Chitin is a basic polymer composed of N-acetyl-glucosamine and N-glucosamine units connected by β-(1–4) bonds [11]. It exists in three crystalline allomorphs: α, β, and γ. The most abundant of these forms in nature is α-chitin, which is found in crustaceans [12,13]. Crustacean shell waste is used commercially as a chitin source [12,13,14]. The α-chitin chains are organised into sheets with numerous hydrogen bonds holding the sheets tightly together [11,12]. Extensive hydrogen bonding between the amide groups in the crystalline structure of α chitin results in problematic dissolution of chitin in many solvents. Chitin is hydrophobic and insoluble in most organic solvents [11,12]. Extraction procedures require elevated temperatures and the use of strong acids and bases.

### 3.2. Extraction of Chitin

The chemical method of chitin extraction is a two-step process of demineralisation and deproteinisation of the crustacean shells, as shown in Figure 1. Demineralisation generally involves a strong acid such as hydrochloric acid (HCl) solution to dissolve CaCO_3_. This is followed by deproteinisation which utilises an alkali such as sodium hydroxide (NaOH) solution to dissolve proteins [2].

In general, DES are comprised of two, or sometimes three, inexpensive and safe components, which bind together through hydrogen bond interactions, producing a mixture with a lower melting point than those of the original components. This reaction requires both a hydrogen bond acceptor (HBA) and a hydrogen bond donor (HBD) [4]. For chitin extraction, choline chloride (ChCl) is commonly used as an HBA, while HBDs include lactic acid, malonic acid, and citric acid [15]. DES facilitate the removal of both CaCO_3_ and protein in a single step as illustrated in Figure 1.

### 3.3. Measurement of Chitin Purity

Chitin purity can be considered as the weight percentage of extracted material which is neither CaCO_3_ nor protein. Feng et al. [3] calculated the percentage purity using the following formula: Purity% = 100 − CaCO_3_% − protein%. However, elements such as chlorine (Cl), phosphorus (P), and silicon (Si) may also be present in addition to the CaCO_3_ and protein [4]. Saravana et al. [2] demonstrated that the quantity of moisture (the percentage of extracted chitin product which is water) in the sample may also affect chitin purity calculations.

The amount of CaCO_3_ remaining in the chitin sample can be determined by heating at a high temperature in a muffle furnace until the sample is at constant weight. The remnant ash is CaCO_3_, and its weight can be calculated as a percentage of the original sample weight. This method is often referred to as an ash test [9]. Protein assays are widely used to determine the protein concentrations in a variety of research applications. The Bradford assay [9] and Lowry’s assay [2] are commonly used for the quantification of protein within chitin samples. The quick and simple nature of both colorimetric methods for protein determination means that they are suitable for this application [16].

The chitin samples were characterised and represented under up to five headings, as shown in Table 1, Table 2 and Table 3. Firstly, the CaCO_3_ (%) represents the percentage of extracted sample which is CaCO_3_ and was measured by an ash test. The protein (%) refers to the amount of protein in the chitin extracts as a percentage of the chitin extract and was measured by a protein assay. The heading titled other (%) represents all materials in the chitin extract which are not chitin, protein, or CaCO_3_. This includes the quantity of moisture and additional impurities including elements Cl, P, and Si. The yield refers to the weight of the extracted material as a percentage of the starting material (crustacean shell), calculated by the following equation.

Yield (%) = (Extracted material (g)/Starting material (g)) × 100

The quantity of chitin in chitin samples can be referred to as the purity and this value can be calculated from the following equation.

Chitin or Purity (%) = 100 − (CaCO_3_ (%) + protein (%) + other (%))

This value is referred to as chitin when discussing the quantity of chitin in crustacean shells (Table 2) or purity when discussing a chitin extract (Table 1 and Table 3), because this value is ideally close to 100% for chitin extracts.

**Table 1 molecules-26-07603-t001:** The purity, yield, and constituents of chitin extracted using the chemical method. Species habitat was identified through searches of the Global Biodiversity Information [17], the Food and Agriculture Organisation of the United Nations Fact sheets [18,19,20] and the Marine Life Information Network [21].

Marine Habitat	Chitin Source	CaCO_3_ (%)	Protein (%)	Other (%)	Purity (%)	Yield (%)	Reference
All seas except polar	Lobster shells (*Nephropidae*)	0.39 ± 0.23	2.22 ± 0.24	3.93 ± 0.09	93	17.21 ± 0.28	[22]
All seas except polar	Lobster shells (*Nephropidae*)	0.30 ± 0.20	2.90 ± 0.25	4.17 ± 0.03	93	16.53 ± 2.35	[1]
Indian and North Pacific Ocean	Shrimp shells (*Marsupenaeus japonicus*)	0.45 ± 0.10	1.13 ± 0.01	1.32 ± 0.00	97	16.08 ± 0.57	[2]
Eastern Pacific Ocean	Shrimp shells (*Litopenaeus vannamei*)	0.1	0.95	-	-	-	[23]
Eastern Pacific Ocean	Shrimp shells (*Litopenaeus vannamei*)	0.2	0.92	-	-	-	[23]

**Table 2 molecules-26-07603-t002:** The proportions of the constituents in a variety of marine crustacean shell species.

Marine Habitat	Source	Chitin (%)	CaCO_3_ (%)	Protein (%)	Other (%)	Reference
Marine: Eastern Atlantic Ocean and Mediterranean Sea	Lobster shells (*Homarus*)	60–75	-	-	-	[24]
Marine: Eastern Atlantic Ocean and Mediterranean Sea	Lobster shells (*Nephron norvegicus*)	69.8	-	-	-	[24]
	Shrimp shells (species unknown, Egypt)	36.43	32.46	32.77	-	[12]
Marine: Indian and Pacific Ocean	Shrimp shells (*Solenocera crassicornis*)	35.8	56.1	8.1	-	[3]
Atlantic East and West coasts	Shrimp shells (*Parapenaeus longirostris*)	26.98	25.06	29.23	18.73	[12]
Marine: Western Atlantic and North Pacific	Crab shells (*Chionoecetes*)	26.6	-	-	-	[24]
Marine: All seas except polar	Lobster shells (*Nephropidae*) China	26.23	40.64	25.83	7.3	[1]
Marine: All seas except polar	Lobster shells (*Nephropidae*) China	26.23	40.64	25.83	7.3	[23]
Marine: Antarctic	Krill shells (*Euphausia superba*)	24	-	-	-	[24]
Marine: Western Atlantic	Shrimp shells (*Penaeus durarum*)	23.72	42.26	34.02	-	[25]
Marine: Eastern Atlantic Ocean	Shrimp shells (*Penaeus aztecus*)	21.53	48.97	29.5	-	[25]
Marine: Indian and North Pacific Oceans	Japanese tiger prawn (*Marsupenaeus japonicus*)	19.21	31.76	36.47	12.56	[2]
Marine: Eastern Atlantic Ocean and Mediterranean Sea	Shrimp shells (*Crangon crangon*)	17.8	-	-	-	[24]
	Crab shells (species unknown, Egypt)	16.73	66.58	16.68	-	[25]

**Table 3 molecules-26-07603-t003:** The purity, yield, and constituents of extracted chitin using DES-mediated methods.

Marine Habitat	Chitin Source	DES Composition [Molar Ratio]	Shell: Solvent Ratio	Temp °C	Time (Hours)	CaCO_3_ (%)	Protein (%)	Other (%)	Yield (%)	Purity (%)	MW (kDA)	DA (%)	Reference
Coastal mud shrimp	Shrimp shells (*Solenocera crassicornis*)	Choline chloride: Malonic acid [1:40]	1:20	150	3	0.3–0.4	0.5–0.6	-	4.9 ± 1	99.1 ± 0.1		61	[3]
Coastal mud shrimp	Shrimp shells (*Solenocera crassicornis*)	Choline chloride: Malonic acid [1:2]	1:20	150	3	0.6–0.7	0.7–0.8	-	13.2 ± 1.1	98.6 ± 0.2	312	46	[3]
Indian and North Pacific Oceans	Japanese tiger prawn (*Marsupenaeus japonicus*)	Choline chloride: Malonic acid [1:40]	1:40	80	2	0.74	0.74 ± 0.02	1.53 ± 0.02	3.72 ± 0.05	23.86 ± 0.07			[2]
North Atlantic Ocean	Shrimp shells (*Pandalus borealis*)	Choline chloride: Malonic acid [1:1]	1:20	70	3	0.56	0.98	0.46	19–20	98 ± 1			[4]
All seas except polar	Lobster shells (*Nephropidae*)	Choline chloride: Malonic acid [1:2]	1:10	50	2	0.21 ± 0.31	1.81 ± 0.14	4.12 ± 0.21	22.21 ± 0.27	93	312	94.33	[22]
All seas except polar	Lobster shells (*Nephropidae*)	Choline chloride: Malonic acid [1:2]	1:10	70	2	0.34 ± 0.22	1.77 ± 0.22	3.88 ± 0.11	21.01 ± 0.23	93	278	94.21	[22]
All seas except polar	Lobster shells (*Nephropidae*)	Choline chloride: Malonic acid [1:2]	1:10	100	2	0.24 ± 0.16	1.75 ± 0.17	4.11 ± 0.23	19.01 ± 0.24	93	199	95.05	[22]
Indian and North Pacific Oceans	Japanese tiger prawn (*Marsupenaeus japonicus*)	Choline chloride: Malic acid [1:2]	1:40	80	2	1.44 ± 0.01	3.59 ± 0.02	1.92 ± 0.01	25.00 ± 0.60	93			[2]
Indian and North Pacific Oceans	Japanese tiger prawn (*Marsupenaeus japonicus*)	Choline chloride: Citric acid [1:2]	1:40	80	2	1.18 ± 0.01	8.37 ± 0.05	2.32 ± 0.06	25.18 ± 0.38	88			[2]
Indian and North Pacific Oceans	Japanese tiger prawn (*Marsupenaeus japonicus*)	Choline chloride: Malonic acid [1:2]	1:40	80	2	3.60 ± 0.14	13.05 ± 0.20	1.25 ± 0.04	25.22 ± 0.90	82			[2]
Indian and North Pacific Oceans	Japanese tiger prawn (*Marsupenaeus japonicus*)	Choline chloride: Urea [1:2]	1:40	80	2	41.01 ± 1.80	15.34 ± 0.18	4.56 ± 0.02	50.54 ± 1.07	39			[2]
Indian and North Pacific Oceans	Japanese tiger prawn (*Marsupenaeus japonicus*)	Choline chloride: Ethylene glycol [1:2]	1:40	80	2	44.34 ± 3.40	13.50 ± 0.12	4.42 ± 0.04	52.45 ± 2.01	38			[2]
Indian and North Pacific Oceans	Japanese tiger prawn (*Marsupenaeus japonicus*)	Choline chloride: 1,6-Hexanediol [1:2]	1:40	80	2	46.19 ± 1.90	16.14 ± 0.10	6.42 ± 0.09	52.55 ± 0.70	31			[2]

### 3.4. Chitin Purity

There have been numerous reports of chitin extraction from crustacean shells using the chemical method, many of which are represented in Table 1 [1,2,22,23]. Typically, this is a two-step process where shells are agitated in 10% *w*/*v* HCl for 2–4 h to remove CaCO_3_. This is followed by agitation and heating (60–100 °C) with 10% (*w*/*w*) NaOH to remove proteins (Figure 1). Trung et al. [23] extracted high-purity chitin from shrimp shells (*Litopenaeus vannamei*) using variations of the chemical method. The lowest quantity of CaCO_3_ in this study [23] was 0.1% while the quantity of protein for the same sample was 0.95%; the lowest quantity of protein observed was 0.92% while the quantity of CaCO_3_ for the same sample was 0.2%. This latter study focused on chitin extraction using the chemical method and demonstrates very high-purity chitin.

It is worth pointing out that not all chemical-mediated extraction methods result in high-purity chitin. Saravana et al. [2] extracted chitin from shrimp shells (*Marsupenaeus japonicus*) and reported the CaCO_3_ and protein quantities of extracted chitin using the chemical method and various DES formulations. When Saravana et al. [2] employed the chemical method, chitin with 0.45 ± 0.10% CaCO_3_ and 1.13 ± 0.01% protein was extracted. The most promising DES combination utilised for chitin extraction in this study was ChCl and malonic acid which resulted in chitin with 0.74 ± 0.02% CaCO_3_ and 1.53 ± 0.02% protein. Some formulations of DES were not successful in extracting high-purity chitin; for example, a DES combination containing ChCl and glycerol had a high quantity of CaCO_3_ (42.93 ± 2.92%) and protein (22.38 ± 0.24%).

A variety of studies on DES-mediated extractions are represented in Table 3. Different DES formulations influenced the level of CaCO_3_ and protein remaining after extraction, some formulations resulted in increased CaCO_3_ removal, while others better facilitated protein removal. For example, Saravana et al. [2] utilised a DES combination of ChCl and citric acid and produced chitin with a low quantity of CaCO_3_ (1.18 ± 0.01%) but a high quantity of protein (8.37 ± 0.05%). Saravana et al. [2] also investigated ChCl and malic acid (1.44 ± 0.01% CaCO_3_ and 3.59 ± 0.02% protein) and ChCl and L-(+)-tartaric acid (3.60 ± 0.14% CaCO_3_ and 13.05 ± 0.20% protein). In contrast, a DES combination of ChCl and ethylene glycol showed greater deproteinisation ability than demineralisation ability, resulting in chitin containing 13.50 ± 0.12% protein and 44.34 ± 3.40% CaCO_3_. Other DES combinations showed similar results: ChCl and urea (41.01 ± 1.80% CaCO_3_ and 15.34 ± 0.18% protein) and ChCl and 1,6-hexanediol (46.19 ± 1.90% CaCO_3_ and 16.14 ± 0.10% protein) [2].

Saravana et al. [2] reported significant variability in the quantity of moisture in DES-extracted chitin samples with moisture values of between 1.25 ± 0.04% and 6.42 ± 0.09%. Saravana et al. [2] also reported that chitin extracted using the chemical method had 1.32 ± 0.00% moisture. Zhu et al. [1] reported moisture quantities of 4.17 ± 0.03% for chemically extracted chitin, more than three times the percentage of moisture measured in chitin that was chemically extracted by Saravana et al. [2]. With such variability in the quantity of moisture in chitin samples, it is an important factor for consideration when comparing chitin purity. Inaccurate chitin purities could be reported if the quantity of moisture was not included in calculations required to determine the quantity of chitin.

A DES combination of ChCl and malonic acid was utilised by Hong et al. [22] who extracted chitin from lobster shells (species unknown, China). A chitin purity of 93% was reported with 0.21 ± 0.31% CaCO_3_, 1.81 ± 0.14% protein, and 4.12 ± 0.21% moisture [22]. Zhu et al. [1] also utilised a DES combination of ChCl and malonic acid when extracting chitin from lobster shells (species unknown, China). The resultant chitin contained 0.30 ± 0.15% CaCO_3_, 2.02 ± 0.24% protein, and 4.21 ± 0.02% moisture [1]. Upon utilisation of the chemical method, Zhu et al. [1] reported chitin with 0.30 ± 0.20% CaCO_3_, 2.90 ± 0.25% protein, and 4.17 ± 0.03% moisture. Thus, the purity of the chitin that resulted from the utilisation of ChCl and malonic acid was superior to that observed for the chemical method when lobster shells (species unknown, China) were utilised, as demonstrated by both Zhu et al. [1] and Hong et al. [22].

Bradić et al. [4] extracted chitin from shrimp shells (*Pandalus borealis*) with a purity of 98 ± 1% using a DES formulation of ChCl and lactic acid. The final chitin product contained 0.56% CaCO_3_ and 0.98% protein. The authors of this study reported that additional impurities including elements Cl, P, and Si accounted for the remaining 0.46% of the final product, following the DES-mediated extraction method. Feng et al. [3] extracted chitin from shrimp shells (*Solenocera crassicornis*) with various DES combinations. The highest purity of chitin, 99.1 ± 0.1%, was generated using a combination of ChCl and malic acid. Feng et al. [3] reported that chitin extraction occurred optimally in terms of yield and other process parameters when a combination of ChCl and malic acid was utilised, and the chitin purity was 98.6 ± 0.2%. The basis for this conclusion will be discussed in further sections.

To summarise, high-purity chitin can be extracted by the chemical method, under the appropriate conditions. The DES-mediated method can also extract high-purity chitin, under appropriate conditions as demonstrated by several studies, showing the potential for DES in industrial applications. Feng et al. [3] extracted chitin with 99.1 ± 0.1% purity, which exceeds that of any chemical method extraction discussed. However, there are other considerations to address including yield, and quality parameters.

## 4. The Conditions of the Extraction Process

### 4.1. Duration, Temperature, and Solvent Concentration

Variations in processing conditions including reaction time, solvent concentration, preparation and pre-treatment steps, and the type of crustacean shell utilised can affect the purity of the extracted chitin. Furthermore, the solvents required for demineralisation and deproteinisation can also affect the resultant chitin. The influence of the extraction parameters will be discussed with reference to existing literature. Firstly, the process parameters that underpin the chemical method can influence the success of the associated method. Trung et al. [23] demineralised pre-treated, wet shrimp shells (*Litopenaeus vannamei*) at room temperature (28–32 °C) for 12 h in 0.80 M HCl, and deproteinised at room temperature for 24 h in 0.75 M NaOH solution. Variations in the demineralisation and deproteinisation temperatures and incubation durations resulted in different chitin purities, as shown in Table 1.

Similar to the chemical method, variation in chitin extracts from DES-mediated processes can occur due to modification of the parameters employed in the extraction strategy. Firstly, the type of DES can affect the resultant chitin. DES combinations can be varied while other parameters are maintained constant to determine the effect of the variable. For example, Saravana et al. [2] extracted all chitin at 80 °C for 2 h at a ratio of 25 mL of solvent per gram of shrimp shell (*Marsupenaeus japonicus*). The only variable investigated by Saravana et al. [2] was the DES combination. This variable alone can have an immense impact on the purity of chitin extracted, as shown in Table 3 where chitin extracts from the latter study were analysed.

However, the DES combination is not the only parameter which can change within the DES-mediated extraction process; parameters including the duration and temperature of the extraction process as well as the ratio of crustacean shell to solvent can also be varied. A study performed by Bradić et al. [4] demonstrated the effects of varying some of these parameters when extracting chitin using a range of temperatures between 60 and 90 °C, for durations of 3 or 6 h. The amount of shrimp shells (*Pandalus borealis*) also varied, ranging from 1 to 2 g in 50 g of DES [4]. Additionally, Zhu et al. [1] extracted chitin at different temperatures (50 to 110 °C) and durations (2, 4, or 6 h) depending on the type of DES utilised.

Ideally, the lowest possible volume of DES should be utilised to avoid unnecessary waste. However, as demonstrated by Feng et al. [3], the ratio of crustacean shell to DES affects chitin purity. When the mass ratio of shrimp shell (*Solenocera crassicornis*) to DES varied from 1:10 up to 1:50, with all other parameters remaining constant, the purity increased to 99.1 ± 0.1% at a mass ratio of 1:40. By doubling the mass of solvent (increase from 1:10 to 1:20 shell to DES mass ratio), the purity increased significantly from 95.8 ± 0.4% to 98.6 ± 0.2%. Achieving the balance between high purity and using the least amount of DES possible reduces waste and improves the environmentally friendly nature of the process. The latter study [3] also employed high temperatures of 150 °C which may be unnecessarily high considering that chitin extraction has been achieved with the utilisation of lower temperatures. For example, Zhu et al. [1] employed a lower extraction temperature of 50°C and the resultant chitin contained 0.30 ± 0.15% CaCO_3_ and 2.02 ± 0.24% protein. Finding the optimum balance between high purity and lower temperature would improve the efficiency of the extraction process.

### 4.2. The Importance of Crustacean Shell Type

The type of crustacean shell utilised for an extraction process may affect the purity and the quality of the extracted chitin. Currently, the most common source of commercial chitin is shrimp shells [25]. However, since this polymer can be sourced in a wide variety of crustacean shells, investigation into the natural chitin quantity of other crustacean shell species is worthwhile. The natural proportions of chitin, CaCO_3_, protein, and other components such as moisture from a variety of crustacean shells species are shown in Table 2.

A low quantity of chitin of 16.73% in crab shell (species unknown, Egypt) with a high quantity of CaCO_3_ (66.58%) and protein (16.68%) was reported by Abdou, Nagy, and Elsabee [24]. A higher quantity of chitin (36.43%) in shrimp shell (species unknown, Egypt) with 32.46% CaCO_3_ and 32.77% protein was reported by El Knidri et al. [26]. The quantity of chitin was reported to be as high as 69.80% in lobster shells (*Nephron norvegicus*) [27]. These figures demonstrate the variability in the natural composition of crustacean shells between species. Shells with a naturally higher prevalence of chitin, such as lobster shell (*Nephron norvegicus*), can generate a greater yield of chitin per quantity of raw material (crustacean shell). Furthermore, the proportions of CaCO_3_ and protein must be taken into consideration when developing the DES-mediated extraction process. The demineralisation and deproteinisation abilities vary depending on the DES employed as discussed previously in the investigation carried out by Saravana et al. [2]. Therefore, crustacean shells with higher proportions of CaCO_3_ would be suited to those DES with higher demineralisation abilities than deproteinisation abilities. Similarly, crustacean shells with higher proportions of protein would be suited to those DES with better deproteinisation abilities. It should also be considered that the parameters utilised in the DES-mediated extraction process can enhance the demineralisation and deproteinisation abilities of DES. An investigation on the parameters of the extraction process could determine the optimum chitin extraction method for each species of crustacean shell.

Zhu et al. [1] and Hong et al. [22] extracted chitin from lobster shells (unknown species, China). Although the species of lobster shell utilised was unknown for both studies [1,22], the proportions of chitin, CaCO_3_, and protein in both lobster shells were identical (26.23% chitin, 40.64% CaCO_3_, and 25.83% protein), as shown in Table 3. These identical percentages suggest that the same species of lobster shell was utilised by Zhu et al. [1] and Hong et al. [22]. A DES combination of ChCl and malonic acid was employed for both studies at 50 °C for 2 h. Some differences in the extraction process were observed; for example, the shell to DES mass ratio during extraction and the washing of the extracted chitin. However, similar chitin purity results were achieved as shown in Table 3.

The proportions of crustacean shell constituents vary greatly among different species, and this may have an impact on the success of the extraction process. The conditions of both the chemical method and the DES-mediated method may need to be altered depending on the type of crustacean shell undergoing chitin extraction. Further research is required to determine the level of impact that the species of shell has on the extraction process and the final chitin product.

## 5. Further Considerations

### 5.1. Yield

The yield of a product is calculated by the mass difference of crustacean shell and extracted chitin; this value expresses the extracted chitin as a percentage of the crustacean shell weight [2]. A high yield in combination with a high-purity value is favourable as this would imply that less crustacean shell is required to extract a desired amount of chitin. If the extracted chitin is highly pure, the yield should be equal or close to the percentage of chitin in the original crustacean shell. Examples of the constituents of crustacean shells are shown in Table 2. If we compare Table 1 and Table 2 to Table 3 for the appropriate shell species, we can determine approximately how much, if any, chitin was lost during the chemical method or the DES-mediated extraction process.

#### 5.1.1. Losses Purity Is Not the Only Considerationuring Extraction

Zhu et al. [1] reported that lobster shell (species unknown, China) contained 26.23% chitin. The weight of the starting material was 10.00 g for all extractions carried out by Zhu et al. [1]; approximately 2.623 g of chitin was available for extraction. Upon extraction from lobster shell (species unknown, China), the chitin yield was reported at 16.53 ± 2.35% (1.653 g) from the chemical method and at 16.19 ± 2.50% (1.619 g) from the DES-mediated method. The chitin extracts contained small amounts of protein and CaCO_3_ and therefore, less than the reported quantities (1.653 g and 1.619 g) were in fact chitin. At least 1 g of chitin was lost during both extraction methods [1]. Similarly, Hong et al. [22] reported that lobster shell (species unknown, China) contained 26.23% chitin, but the weight of the starting material was not reported. Upon extraction from lobster shell (species unknown, China), the chitin yield was reported at 17.21 ± 0.28% from the chemical method and at 22.21 ± 0.26% from the DES-mediated method. The chitin extracts contained small amounts of protein and CaCO_3_ and therefore, less than the reported yields (17.21 ± 0.28% and 22.21 ± 0.26%) were in fact chitin. However, more chitin was lost during the chemical method than during the DES-mediated method as suggested from the difference in chitin yield between the methods in the study carried out by Hong et al. [22]. Zhu et al. [1] did not report the purity of the chitin extracted by the DES-mediated process while 0.30 ± 0.15%, 2.02 ± 0.24%, and 4.21 ± 0.02% of CaCO_3_, protein, and moisture were reported, respectively. Hong et al. [22] reported the purity of the DES-extracted chitin at 93% while also reporting the quantities of CaCO_3_ (0.21 ± 0.31%), protein (1.81 ± 0.14%), and moisture (4.12 ± 0.21%) present. Both studies extracted chitin of similar purity but when Hong et al. [22] extracted chitin by a DES-mediated process from the same species of lobster shell, a higher yield of 22.21 ± 0.26% was observed, suggesting that losses due to the extraction process can be minimised.

#### 5.1.2. Differences in Chitin Content in Shells

The chitin content of crustacean shells and the resultant chitin yield can be considered when determining how much, if any, chitin was lost during the extraction processes. Saravana et al. [2] determined that shrimp shells (*Marsupenaeus japonicus*) contain 19.21% chitin; a chitin yield was reported for both the chemical method (16.08 ± 0.57%) and a DES-mediated method (23.86 ± 0.07%) when these shrimp shells were used as starting material. The chitin yield appears higher than the chitin quantity of the shrimp shells; however, when factoring in the quantity of moisture in the shrimp shells (12.56 ± 1.09%) compared with the extracted chitin (3.72 ± 0.05%), the difference is justified. Clearly, chitin had been destroyed or removed during the chemical process, but less, or no chitin had been destroyed or removed during the DES-mediated process, as the high yield suggests. Although there is a significant difference in yield, the chemical method reported purer chitin, as discussed previously.

#### 5.1.3. Purity Is Not the Only Consideration

Feng et al. [3] extracted chitin with a purity value of 99.1 ± 0.1% from shrimp shell (*Solenocera crassicornis*) using a DES combination of ChCl and malic acid. However, a very low yield of only 4.9 ± 1% was observed. Feng et al. [3] reported that shrimp shell (*Solenocera crassicornis*) contains 35.8% chitin, suggesting that a large quantity of chitin had been destroyed or removed during the process. Additionally, the environmentally friendly nature of this process is questionable due to the extreme conditions, including high temperatures of 150 °C and an unusually high shell/DES ratio of 1:40. It must be considered that a lower chitin purity may be desirable if the environmentally friendly nature of the process is to be maintained and a high yield observed. This study by Feng et al. [3] demonstrates how chitin purity is not the only important factor when deciding on an extraction process.

### 5.2. Advantages and Disadvantages of the Chemical and DES-Mediated Processes

Many different strategies can be used to extract chitin from crustacean shells and there are multifarious advantages and limitations associated with each of these approaches. Table 4 sets out some of the advantages and disadvantages of the two processes. It is worth noting that while DES-mediated extraction shows significant potential for future development, aqueous chemical extraction has reached the limits of efficiency possible with this approach. 

### 5.3. Economic Potential of Chitin

A recent review of the economic potential of chitin and chitosan by Oyatogun et al., reported in 2015, that the annual global demand for chitin was greater than 60,000 ton, exceeding production of 28,000 ton; these data were based on a global report published in 2015 [30]. The global chitin and chitosan market was valued at USD 2 billion in 2016 with forecasts predicting growth to USD 4.2 billion by 2021 [30]. Projected estimates of the global market for chitin and its derivatives including chitosan were targeted to reach USD 6.38 billion by 2024 to meet increased demands of end-use industries [30]. Cosmetic and pharmaceutical industries are projected to increase chitin/chitosan consumption by 18% in the near future [30].

### 5.4. Application of Chitin and Chitin Derivatives Such as Chitosan

Chitin and its derivatives (chitosan being the most extensively studied thus far), have many applications in diverse industries, which are summarised in Table 5. Chitosan is a polymer composed of β-(1–4)-linked D-glucosamine units. Properties including biocompatibility, biodegradability, nontoxicity, increased adsorption capacity, and free radical scavenging mean that chitosan is a suitable polymer for many biomedical applications. Chitin and its derivatives can be prepared into films, fibres, sponges, beads, powder, gel, and solutions.

A highly significant application of chitosan is in disease treatment. Chitosan as a dietary supplement has been used as a reno-protective, pre-dialysis treatment for chronic renal failure with efficacy in the removal of uremic toxins and cholesterol [41]. Surface-deacetylated chitin nanofibers (SDACNFs) can be used as multifunctional nanomaterials for drug delivery of poorly water-soluble drugs [41]. Furthermore, SDACNFs used for drug delivery offer additional antioxidant benefits in the treatment of inflammatory bowel disease (Crohn’s and ulcerative colitis) [41].

## 6. Conclusions and Recommendations

This review reveals that DES-mediated extraction of chitin from crustacean shells offers several advantages compared to the chemical method. The former can facilitate the extraction of higher yields of chitin from a variety of crustacean shells with higher purity (99.1% purity). Despite the latter advantages, associated extraction procedures require further consideration; differences in the properties of shells from different species will likely require species-specific DES to extract chitin. Furthermore, the temperatures employed (50–150 °C) in the most successful protocols require significant energy expenditure. The possibility of solvent recycling to maximise the environmentally friendly nature of DES-mediated extractions has not been fully explored. Although significant research has been conducted to investigate the effects of different DES formulations, many of these studies have focused on choline chloride and malonic acid. The exploration of alternative DES formulations may improve the potential of DES-mediated extractions.

## 7. Future Prospects

The chitin polymer has significant potential in many industrial applications. Chitosan and other chitin derivatives will continue to be utilised in water decontamination. Chitosan and chitin-derived food packaging films could represent an ecological solution to food packaging. Over 100 clinical trials involving chitosan have been registered; 25 of these were registered in 2020 or after [42]. These trials are investigating the effectiveness of chitosan as: a dietary supplement for obesity; a treatment for eye disease; a cancer treatment; a coating for facemasks to reduce COVID-19 infections; a cartilage repair component; and a component in multifarious wound and tissue repair products. Should effectiveness be proven in these clinical trials, the market demand for chitin will continue to rise. Thousands of patents have been filed in 2021 based on inventions related to chitosan modifications and applications. The utilisation of chitin could be further advanced by synthesising derivatives to serve as alternatives to chitosan and tailoring molecules to suit the requisite applications. Finally, future sustainable utilisation of this renewable and abundant biopolymer would benefit from a green chemistry extraction procedure.

## Figures and Tables

**Figure 1 molecules-26-07603-f001:**
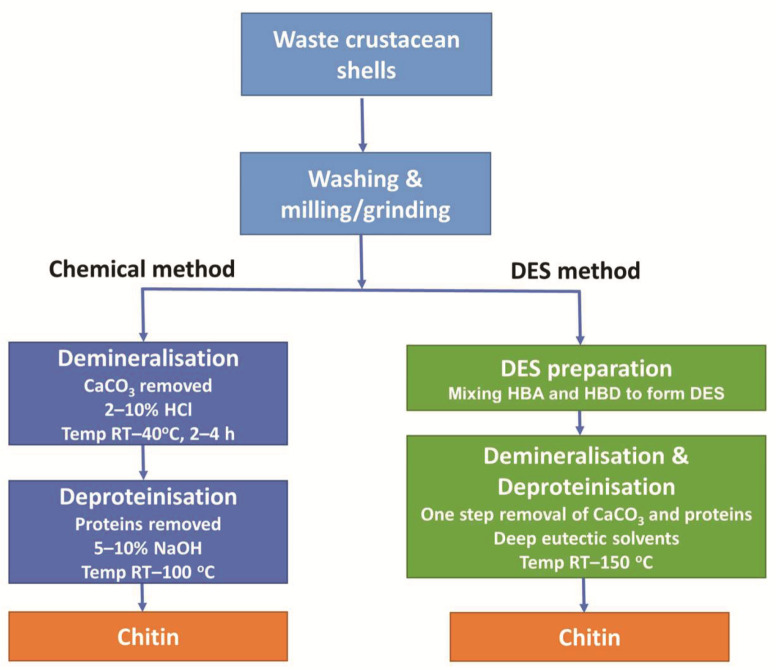
Comparison of steps utilised in chemical and DES-mediated extraction procedures to extract chitin from crustacean shells.

**Table 4 molecules-26-07603-t004:** Advantages and disadvantages of DES- and chemical-mediated extractions.

DES Extraction	Chemical Extraction	Reference
Single step for simultaneous removal of protein and minerals	Two or three steps required to remove protein and minerals	[2]
Solvent recycling possible	Large quantities of waste generated, high cost of treatment	[4]
Calcium carbonate recoverable	Calcium carbonate lost to waste stream	[28]
Proteins and amino acids recoverable	Protein lost to waste stream	[27]
Molecular weight of chitin conserved	Molecular weight reduced during processing	[29]
No deacetylation	Some deacetylation unavoidable	[29]
High solvent viscosity causes difficulty at large scale	Low viscosity suited to large scale applications	[15]

**Table 5 molecules-26-07603-t005:** Properties of chitin/chitin derivatives and applications in different industries (reviewed extensively in [31] and summarised here).

Industry	Favourable Properties	Use	Reference
Cosmetics	Biocompatibility, non-toxic, high thermostability, good solubility in acidic media and cosmetic bases, stability in pH range, neutral or pleasant odour with low volatility.	Component of the following products: Shampoos, rinses, colourants, hair lotions, spray, and tonics. Sunscreens, moisturiser foundation, eyeshadow, lipstick, cleansing materials, and bath agent, toothpaste, mouthwashes, and chewing gum as a dental filler.	[32]
Water	Flocculating, and negative charge (chelating agent).	Wastewater treatment for removing heavy metal ions and decontamination.	[32,33]
Paper industry	Structural integrity.	Production of recycled paper and packaging material.	[32]
Textile industry	Structural integrity.	Dye removal.	[32]
Food industry	Adsorbent and antioxidant.	Nonabsorbable carrier, thickener, and gelling agent, emulsifying agent, antioxidant agent.	[32,34]
Food industry	Ability to form films, antimicrobial activity.	Semipermeable, tough, long-lasting, flexible films, used as food wrapping.	[34,35,36]
Agriculture	Antifungal	Antifungal treatment for plant pests. Fruit preservative. Controlled delivery of fertilisers, pesticides, and insecticides.	[37]
Aquaculture	Immunostimulant	Aquaculture feed.	[38]
Photography		Fixing agent.	[32]
Medicine: Tissue engineering	Nontoxicity, biocompatibility, biodegradability, structural integrity, mechanical properties.	Repair, replacement, maintenance, or enhancement of the function of a particular tissue or organ. Bone repair.	[39,40]
Medicine: Wound healing/wound dressing	Nontoxicity, biocompatibility, biodegradability, structural integrity, film formation.	Semipermeable to oxygen, tough, wound dressings for burns etc.	[39,40]
Medicine: Drug delivery	Adsorbable and nontoxicity.	Slow release of drugs, for more efficient drug delivery.	[39,40]

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
