# Peer review of "A Comparative Analysis of Conventional and Deep Eutectic Solvent (DES)-Mediated Strategies for the Extraction of Chitin from Marine Crustacean Shells"

_molecules, 2021, doi:10.3390/molecules26247603_

Round 1
Reviewer 1 Report
Remarks
As I know, even though the review article but the authors as per guidelines of the editorial policy have to draft their review article by following a methodology and then eventually results and discussion. In this article I do not find such alignment except a thorough review only focusing particular subject with limited scope.
I would suggest the authors to rewrite it by analyzing previous studies and providing them in the form of tables or graphs for better understanding. Moreover, the authors must broaden the scope of the paper by relating to commercial aspects or cost analysis of chitins for a broad readership. Furthermore, the authors must also discuss future emerging trends in this regard.
The manuscript under the current form, although it is very well written from the language point of view, but from readership, contents, scope, and novelty, it is a bit weak.
- I would suggest rewriting the abstract by keeping in mind the incorporation of above-mentioned points.
- Both the headings 2. (Chitin Purity), and sub-heading 2.2 are the same “Chitin Purity” better use appropriate heading to avoid duplication.
- Line 358-385: Better to analyze in the form of a table for better understanding and broad readership.
- Conclusion and recommendations are too long and seem like again a review of the literature. I would suggest the authors to rewrite your findings and recommendations based on such a thorough review of the literature.
Author Response
Please see attached word document

Reviewer 2 Report
While reading the manuscript, I had some questions and comments.
- Supplement the manuscript with a table with information on the physical and chemical properties of chitin isolated from various sources, including molecular weight, viscosity, and more. Compare the properties of chitin of marine and river origin.
- Provide information on the solubility of chitin in different DES.
- Provide data in Table 2 or in an additional table, indicating the DES composition, extraction method, extraction module, extraction time, literature reference.
- Also provide data on extraction by conventional methods. Compare the data obtained in terms of purity, time, yield, biological activity.
- It is known from the literature that the same deep eutectic solvents can be used for the extraction of both lipophilic and hydrophilic compounds by adjusting the proportions of the components. What substances are by-products of DES extraction and conventional methods. How these approaches affect the toxicity of the resulting products.
- What methods are used to purify chitin from DES. What are the prospects for the use of chitin extracts in DES for use in medicine, cosmetology, food. Please provide data to support the use of these extracts without purification.
Author Response
Please see attached word document.

Round 2
Reviewer 1 Report
Remarks
The revised MS is much improved. But, I am still not fully satisfied with the authors, and I would encourage them to incorporate the followings
- The abstract is not so informative, better to rewrite and shorten it. Currently, it seems like an introduction.
- Keywords used in the title are the same as those used as keywords. Both must be different.
- Section 70-77 is providing the methodology for the literature selection criterion. Instead of providing methodology here better to shift under the materials and methods section.
- Line 398: reported in 2105?
- Overall, I found that the text under some of the headings is too much and fully packed with text. I would suggest splitting them into sub-headings and where possible convert into infographics/graphs etc., to grab the reader’s attention.
Author Response
See attached document

Reviewer 2 Report
I have no more questions.
Author Response
Thank you for reviewing the article. Some final changes have been made as described in the attached file.
